# Mast Cell and Astrocyte Hemichannels and Their Role in Alzheimer’s Disease, ALS, and Harmful Stress Conditions

**DOI:** 10.3390/ijms22041924

**Published:** 2021-02-15

**Authors:** Paloma A. Harcha, Polett Garcés, Cristian Arredondo, Germán Fernández, Juan C. Sáez, Brigitte van Zundert

**Affiliations:** 1Instituto de Neurociencia, Centro Interdisciplinario de Neurociencia de Valparaíso, Valparaíso 2381850, Chile; 2Institute of Biomedical Sciences (ICB), Faculty of Medicine & Faculty of Life Sciences, Universidad Andres Bello, Santiago 8370186, Chile; p.garcsgimnez@uandresbello.edu (P.G.); carredor@uc.cl (C.A.); g.fernndezvillalobos@uandresbello.edu (G.F.); 3CARE Biomedical Research Center, Faculty of Biological Sciences, Pontificia Universidad Católica de Chile, Santiago 8330005, Chile; 4Departamento de Fisiología, Facultad de Ciencias Biológicas, Pontificia Universidad Católica de Chile, Santiago 8331150, Chile; 5Department of Neurology, University of Massachusetts Medical School, Worcester, MA 01605, USA

**Keywords:** hemichannels, connexin, pannexin, mast cells, glial cells, inflammation, degranulation, neurodegeneration, pro-inflammatory compounds, gap junction channels

## Abstract

Considered relevant during allergy responses, numerous observations have also identified mast cells (MCs) as critical effectors during the progression and modulation of several neuroinflammatory conditions, including Alzheimer’s disease (AD) and amyotrophic lateral sclerosis (ALS). MC granules contain a plethora of constituents, including growth factors, cytokines, chemokines, and mitogen factors. The release of these bioactive substances from MCs occurs through distinct pathways that are initiated by the activation of specific plasma membrane receptors/channels. Here, we focus on hemichannels (HCs) formed by connexins (Cxs) and pannexins (Panxs) proteins, and we described their contribution to MC degranulation in AD, ALS, and harmful stress conditions. Cx/Panx HCs are also expressed by astrocytes and are likely involved in the release of critical toxic amounts of soluble factors—such as glutamate, adenosine triphosphate (ATP), complement component 3 derivate C3a, tumor necrosis factor (TNFα), apoliprotein E (ApoE), and certain miRNAs—known to play a role in the pathogenesis of AD, ALS, and other neurodegenerative disorders. We propose that blocking HCs on MCs and glial cells offers a promising novel strategy for ameliorating the progression of neurodegenerative diseases by reducing the release of cytokines and other pro-inflammatory compounds.

## 1. Mast Cell Generalities

Mast cells (MCs) are resident immune cells from vascularized tissues that are closely associated with blood vessels, glia, and neurons. In the central nervous system, MCs are typically found within the dura mater, in the brain side of the blood–brain barrier (BBB), choroid plexus, metencephalon, the thalamic–hypothalamic region, hippocampus, and olfactory bulb [1,2,3,4,5].

In 1878, Paul Ehrlich described in his doctoral thesis aniline dye-positive granular cells (“Mastzellen”), or well-fed cells found in the connective tissue [6]. Nowadays, it is accepted that their main morphologic feature is numerous granules, which are not due to hyper-nutrition but correspond to the storage of pre-formed inflammatory mediators in proteoglycan and protease-enriched granules [7,8]. MCs originate from hematopoietic bone marrow precursors in response to a stem cell factor, which is recognized by the tyrosine kinase receptor CD117 (or c-Kit receptor); promoting growth, survival, and migration. Precursors migrate into the blood to infiltrate peripheral tissues and differentiate at local microenvironments. The maturation dependence of local factors has led to recognizing MCs heterogeneity in mammals, variable cell numbers, size, recognition of damage/pathogens, activation response, sensitivity to inhibitors, granule numbers, and content along the same body. In this context, two subtypes have been historically classified in rodents: connective tissue MCs (found in peritoneal cavity, lungs, and skin) and mucosal MCs (gastrointestinal tract and urinary bladder). Interestingly, MCs found in the brain share both phenotypes, with staining characteristics similar to connective tissue MCs but the ultrastructural appearance and secretory content closer to the mucosal MCs [9]. Independent of the tissue where they reside, rodent MCs share a characteristic expression of the stem cell factor (SCF) receptor CD117, high-affinity serum Immunoglobulin E (IgE) receptor Fc epsilon RI (FcεRI), and histamine inside the secretory granules [7].

MC granules present a diverse secretory content including growth factors, cytokines, chemokines, and mitogen factors [10,11,12]. These mediators can be roughly classified in pre-formed (stored in secretion granules, including amines, proteases, proteoglycans, lysosomal enzymes and cytokines such as tumor necrosis factor (TNFα) and SCF and de novo synthesized mediators (including leukotrienes, prostaglandins, cytokines, chemokines, nitric oxide (NO), and reactive oxygen species (ROS), presenting in both cases different pathway for secretory release (Figure 1). De novo production of cytokines includes constitutive secretion, while pre-formed release implies regulated exocytosis [13,14]. Since MCs are secretory cells highly specialized into the massive release of all granular content in a single event, this process is often referred to as degranulation response [15]. Although in vivo compound exocytosis is the predominant form of regulated exocytosis during an allergic response, the mode for degranulation is dictated by the stimulus [16]. Furthermore, at a cellular level, MCs display a heterogeneous granule content, including association with different vesicle fusion proteins [13,17,18,19] (Figure 1).

As a survey innate immune cells, MCs express a plethora of pathogen and damage-associated molecular pattern receptors, which allow them to be activated by complement fragments (C3a, C4a, C5a), neuropeptides calcitonin gene-related peptide (CGRP), corticotropin-releasing hormone (CRH), neurotensin, substance P, somatostatin) and physical conditions (cold, heat, pressure, stress, vibrations) [12]. Depending on the receptor and the signal pathway recruited, MCs activation can induce the release of their granule content and/or de novo synthesis of inflammatory mediators. Activation through compound 48/80 [26,27], neuropeptides [26,28,29,30], and IgE crosslinking reaction [31,32,33,34,35,36,37] leads to both degranulation and constitutive secretion. Whereas activation without degranulation has been demonstrated upon exposure to lipopolysaccharides (LPS) [38] or polyinosinic:polycytidylic acid (Poly(I:C)) viral-like particles [39,40,41]. Moreover, degranulation without constitutive secretions has been demonstrated after complement peptides activation [42,43,44].

## 2. Hemichannels Overview

Hemichannels (HCs) are non-selective plasma membrane channels permeable to ions, and small molecules such as glucose, adenosine diphosphate (ADP), and adenosine triphosphate (ATP), favoring cellular communication. In mammals, two families of proteins form HCs: connexins (Cxs) and pannexins (Panxs) [45,46,47]. Despite the absence of sharing homology in their amino acidic sequence (<16%), the oligomerization of these transmembrane proteins leads to the formation of functional membrane channels. Some of these channels allow cellular communication via the release of small molecules including signaling molecules such as ATP, a reduced form of nicotinamide adenine dinucleotide (NADH^+^), glutamate, and prostaglandin E_2_ [48,49,50] as well as the uptake of glucose from extracellular media [51,52]. Additionally, molecules released via open HCs coordinate paracrine and autocrine cell responses by activating other receptors in neighboring cells, such as purinergic receptors. ATP and their hydrolysis products, ADP, AMP (adenosine monophosphate), and adenosine, can trigger different responses through P2X, P2Y, and P1 receptors. Notably, these responses can also be propagated to cells lacking those receptors via gap junction channels (GJCs) [53,54]. Gap junctional communication allows direct cell-to-cell diffusional exchange of cytoplasmic ions and small molecules [47].

Cxs compose a family of highly conserved membrane proteins. To date, 21 genes encoding human Cxs have been described and named by the suffix Cx followed by a number that correspond to the approximate molecular mass. Each Cx presents four transmembrane domains, where the amino and carboxyl ends are oriented toward the cytoplasmic face of the cell membrane [55]. An increase of the Cx HC activity can be promoted by mechanical stretch, metabolic stress, neurotransmitter receptor activation, absence of extracellular divalent cations, and increase of cytoplasmic calcium ion (Ca^2+^). In addition, Cxs also form GJCs between astrocytes, oligodendrocytes, and ependymal cells, forming panglial networks [47].

On the other hand, the Panx family members are coded by three genes named PANX 1, 2, and 3 [56,57]. The Panx1 protein, the best-characterized family member, forms a membrane channel that in its fully open stage corresponds to channel through which ATP can be released to the extracellular medium [57,58]. The activity of this channel can be enhanced by an increase of intracellular Ca^2+^ concentration, alkaline pH, high extracellular potassium ion concentration, and extracellular ATP [49,57,59,60,61,62]. In contrast, the activity of this channel is reduced via phosphorylation by protein kinase A (PKA) [63]. Upon prolonged opening of this channel, a high amount of ATP (>1 mM) can be released, which can promote neuronal cytotoxicity through the activation of P2X receptors. The latter causes a sustained intracellular free Ca^2+^ concentration increase, membrane depolarization, and decreased ATP production, leading to neuronal death [57,64,65,66,67,68,69,70,71,72]. However, high extracellular ATP concentration can block the Panx1 HC [73] and, a persistent (more than 15 min) high extracellular ATP concentration causes the down-regulation of the channels, reducing the number of channels available in the cell membrane [74]. In this context, Panx1 HCs directly modulate the initiation and propagation of Ca^2+^ waves, although the channel per se is not permeable to Ca^2+^ [24]. For a complete guide on Cxs and Panxs HCs, particularly in the brain, we recommend reading a recently published article [47].

## 3. Hemichannels on Mast Cell Immunological Function

Classic activation induced by antigen recognition through the IgE–FcεRI crosslinking reaction is a well-known process that leads to rapid degranulation, lipid mediators release, and de novo cytokines synthesis. For the degranulation response, which is induced within seconds, soluble IgEs first bind with high affinity to the multichain immune recognition class receptors FcεRIs. After antigen recognition crosslinking, the aggregation of the protein complex induces the activation of the Src kinases Lyn and Fyn and the recruitment of the ZAP70 tyrosine kinase family member Syk to the receptor complex and its subsequent activation. Phospholipase C (PLCγ) is also activated and hydrolyzes phosphatidylinositol 4,5-bisphosphate (PIP_2_) into inositol 1,4,5-trisphosphate (IP_3_) and diacylglycerol (DAG). IP_3_ promotes the release of Ca^2+^ from intracellular stores and DAG activates protein kinase C (PKC). Thanks to Ca^2+^ extrusion and chelation mechanisms, the depletion of intracellular stores rapidly activates the endoplasmic reticulum Ca^2+^ sensor stromal interaction molecule 1 (STIM1). The latter activates store-operated plasma membrane Ca^2+^ channel calcium release-activated calcium channel protein 1 (ORAI1), which recruits an additional plasma membrane Ca^2+^ permeable channel such as the transient receptor potential channels (TRPs) [75], increasing the membrane permeability to Ca^2+^ and slowly replenishing intracellular stores. This signal is essential for MCs degranulation and chemotaxis [76]. Indeed, during intracellular Ca^2+^ measurement, ovalbumin (OVA) recognition induced a fast-transient peak followed by a persistent increase of the Ca^2+^ signal mediated by ionotropic purinergic P2X receptors activated by ATP released via Panx1 HCs [24].

As recently published, the opening of Panx1 HCs can also be induced after local submembrane Ca^2+^ increase induced by nicotine acetylcholine receptor α7 (nAChR) activation in chromaffin cells [77]. A similar mechanism might operate in MCs. since nicotine receptors are expressed in murine MCs [78,79], although a contradictory modulation of degranulation has been reported [80]. Murine MCs express Cx43, Cx32, and Panx1 [25,81] (Figure 1) and form functional GJCs containing Cx43 with fibroblasts [82,83,84]. An important volume of evidences have characterized cellular events for MCs activation during allergy and neuroinflammatory conditions, but few studies have considered MCs HC-forming proteins during the degranulation process. In this review, we will also include indirect evidences for the potential contribution of MC HCs in different neuroinflammatory conditions. 

## 4. Contribution of Mast Cells Hemichannels on Neuroinflammatory Conditions 

### 4.1. Alzheimer’s Disease

The most prevalent form of dementia in the elderly worldwide is Alzheimer’s disease (AD), which is characterized by an abnormal accumulation of amyloid β plaques and neurofibrillary tangles, neuronal death, and synapse loss. So far, intense work over GJCs and HCs cellular communication at early stages and during the progression in AD models have been extensively revised [47,85,86,87,88,89]. First evidences started with the detection of high Cx43 immunoreactivity at the amyloid plaque’s sites from biopsies of AD patients [90]. Similarly, in the double transgenic mice APPswe/PS1dE9, an enhanced immunoreactivity of Cx43 and Cx30 was found at the amyloid plaques, which was mainly associated to astrocytes; and interestingly, although microglia cells surround the amyloid plaques, Cx43 reactivity was not detected in these cells [91]. Long-term administration with a non-specific HC inhibitor INI-0602, derived from glycyrrhetinic acid, did not altered the amyloid deposition nor the activation marker expression (CD11b and glial fibrillary acidic protein (GFAP)), but it inhibited the excessive glutamate release and reduced memory impairment in AD mice models [92].

Using in vitro approaches, it has been possible to establish that after 24 h of amyloid Aβ 25–35 incubation, cultured cortical microglia release an important amount of ATP and glutamate to the extracellular milieu through a Cx43- and Panx1 HC-dependent mechanism [68]. In contrast, cultured astrocytes only after 72 h of amyloid Aβ 25–35 incubation released glutamate and ATP but mainly via Cx43 HCs [68]. Treating neurons with conditioned medium by astrocytes exposed to peptide amyloid Aβ 25–35 also induced the activation of neuronal Panx1 and Cx36 HCs, promoting neuronal death through a mechanism that involves the activation of neuronal P2X7 and N-methyl-D-aspartate (NMDA) receptors [68]. However, in coronal brain slices, which present a more natural and complex cellular micro environment, shorter incubation times (3 h) with amyloid peptide fragments increased the HC activity of different cells. Particularly, increased HC activity has been detected in pyramidal neurons, astrocytes, and microglia from hippocampus, and the neuronal death depended exclusively on astrocytic Cx43 HCs activation [68], suggesting the involvement of a paracrine cell–cell signaling via HCs. 

Astrocytes located at amyloid plaque foci of hippocampal brain slices from APPswe/PS1dE9 mice, which also presented elevated Cx43 HC activity, unexpectedly show unaltered gap junctional communication [93], contrasting with the reduced gap junctional communication found in cultured astrocytes under inflammatory conditions [51,94]. In this animal model, the astroglial genetic deletion of Cx43 prevented the dye uptake increase, Ca^2+^ influx, ATP, and glutamate release, and it increased mitochondrial superoxide generation and the dystrophic neurites of hippocampal neurons [93]. Interestingly, long-term treatment of APPswe/PS1dE9 mice with boldine, an HC but not a GJC inhibitor, prevented the astrocyte and microglia activation, astrocytic Ca^2+^ signal alteration, and glutamate and ATP release, and it alleviated the hippocampal neuronal suffering [95]. Together, these data highlight the importance of astrocytic HCs activation during the progression of AD.

In brain samples from AD patients, activated tryptase-positive MCs have been found in close association with amyloid plaque lesions [96]. In contrast, in skin and stomach samples, phagocyted amyloid fragments inside MCs were also detected [97]. Indeed, novel experiments have shown that rat peritoneal MCs recognize and phagocyte amyloid fragments (fibrillar 1–40 and 1–42 peptides), inducing histamine secretion that depends on the CD47/β_1_ integrin/G_i_ protein membrane complex [98,99]. In addition to the amyloid beta fragments, the amyloid A protein precursor, called Serum Amyloid A (SAA), is also elevated during inflammation in AD [100,101]. Notably, SAA also induces degranulation, cytokine production (TNF-α and interleukin 1 (IL-1β)) and chemotaxis of human MCs [102,103]. Granule contents, possibly through tryptase and heparin action, induce the degradation of SAA, leading to the formation of protofibrillar intermediates [103], suggesting an active role of MCs in the amyloid deposit formation. In addition, in bone marrow-derived MCs, the acute exposure of amyloid β peptide fragment 25–35 induces the degranulation of histamine via a mechanism that depends on Panx1 HCs [25]. The amyloid activation increases the cell membrane current, dye uptake, and cytoplasmic Ca^2+^ signal, which were reduced by Panx1 HC inhibitors or abrogated by the absent of Panx1 in MCs from Panx1 null mice [25]. Interestingly, only uptake of 2-(4-amidinophenyl)-1H-indole-6-carboxamidine (DAPI) and not ethidium bromide was detected, suggesting that the selectivity of Panx1 HCs depends on the cell condition or the stimulus used to increase the Panx1 HC activity. Moreover, in brain MCs, particularly from the prefrontal cortex, amyloid peptide increases the permeability of both Cx43 and Panx1 channels to ethidium bromide, contributing to degranulation evaluated as histamine release [25]. In APPswe/PS1dE9 mice, brain MC population was drastically increased with respect to control mice, particularly in the hippocampus and cortex [25]. Indeed, the infiltration (and/or proliferation) of MCs in these areas was detected even before the onset of amyloid plaque detection (at 3 months post-partum) [25]. In these animals, prefrontal cortex MCs presented the “basal” uptake of ethidium bromide, which was drastically reduced by Panx1 and Cx43 HC inhibitors, suggesting a simultaneous and reciprocal intervention in the activation mechanism of these channels [25]. An important difference could occur when comparing cells during the initial and chronic inflammatory phases of AD. It has been proposed that during the onset of AD, particularly before amyloid plaque formation, MCs recognize low-soluble amyloid β fragments and migrate to the site where plaque aggregation will occur, in which they secrete inflammatory molecules and eventually compromise the BBB [25]. After activation, MCs release protease gelatinases (metalloproteinases 2 and 9) and vascular endothelial growth factor (VEGF), leading to vascular leakage, leukocyte infiltration, and edema [104,105,106]. As the amyloid plaques matures, it is likely that microglia become reactive, secrete pro-inflammatory cytokines, and release glutamate/ATP through HCs, as it has been shown in vitro [68]. The latter could contribute to activate Cx43 HCs in adjacent microglia and astrocytes during the progression of AD. This condition could be worsen by other associated inflammatory conditions in which MCs activation leads to BBB permeabilization, including cerebral ischemia [106,107,108], traumatic brain injury [109,110], experimental autoimmune encephalomyelitis [111,112], and stress [113].

Targeting MCs activation has been explored, particularly using the MC stabilizer masitinib (AB1010), which is a CD117 tyrosine kinase inhibitor developed by AB Bioscience, S.A. (France). Treatment with masitinib prevents the differentiation, migration, and activation of MCs [114]. Masitinib also targets platelet-derived growth factor receptors, while it weakly affects Lyn, Fyn, and the focal adhesion kinase pathway [114]. Masitinib totally prevents the amyloid-induced dye uptake in bone marrow-derived MCs and brain MCs [25]. Importantly, the potential effect as HC blocker has been ruled out using HeLa cells transfected with Cx43 or Panx1. In this system, the HC activity induced by exposure to divalent cation-free solution or mechanical stress, respectively, was not affected by masitinib [25].

A recent study evaluated the chronic effect of masitinib on APP/PS1dE9 mice progression [115]. Although masitinib treatment did not affect the amyloid plaque load nor the IL-1β concentration in older animals treated daily for almost 2 months, it promoted the recovery of spatial learning performance. Additionally, the decrease of synaptophysin immunoreactivity detected in APP/PS1dE9 mice was completely recovered to normal values after MC depletion or treatment with masitinib, suggesting an additional synaptic protection [115]. Thanks to the AB Bioscience support, the anti-inflammatory effect of masitinib is currently considered for the treatment of diverse diseases in which inflammation is either the cause or the consequence of the condition, including cancer, rheumatoid arthritis, inflammatory bowel disease, asthma, multiple sclerosis, ALS, and AD [116,117,118,119]. A randomized placebo-controlled phase II trial study has revealed that 24 weeks of masitinib administration prevents the cognitive decline in mild to moderate advanced AD patients [116].

### 4.2. Harmful Stress Conditions

Stress typically activates the hypothalamic–pituitary–adrenal axis, leading to the production of catecholamines and glucocorticoids through the release of corticotropin-releasing hormone (CRH). Interestingly, CRH affects the BBB permeability through MCs activation [113,120]. In rats, acute non-traumatic immobilization stress during 30 min promotes dura mater MCs degranulation and increased MC protease I content in the cerebrospinal fluid, which is completely prevented by CRH antiserum [120]. This experimental condition also increases diencephalon, cerebellum, and brainstem BBB extravasation of Technetium gluceptate, which is prevented by the MC stabilizer cromolyn [113].

Through a CRH receptor type 1-mediated response, the cyclic adenosine monophosphate (cAMP) concentration of human MCs increases after 3 min of CRH exposure. Interestingly, the degranulation is not triggered, since neither histamine nor tryptase were detected in the extracellular solution after 30 min of incubation. However, the activation of CRH receptors promotes the proliferation and released of preformed TNF-α [121], which is presumably stored in different granules from those containing histamine or tryptase. In addition, the de novo synthesis of IL-6, IL-8, and TNF-α cytokines was undetectable, but VEGF was produced and released [122]. In addition, recognition of the released VEGF could serve as chemotactic signal promoting microglia migration to the injury sites [123], suggesting a potential cross-talk between MCs and microglia during stress responses [124,125,126].

So far, the HCs contribution during stress-mediated MCs activation has not been extensively studied. However, the HC activity of glial cells has been evaluated in both acute and chronic restrain conditions [127]. In both cases, Panx1 HCs from hippocampal microglia are activated, increasing the ethidium bromide uptake, and chronic restrain stress promoted the strongest response [127]. Similarly, astrocytes and neurons presented high HCs activity dependent on both Cx43 and Panx1 HCs [127]. Interestingly, during chronic stress, the inhibition of NMDA/P2X_7_ receptors prevented the increase in both Panx1 and Cx43 HCs activity, and HCs-mediated glutamate and ATP release [127].

In rats, stress induced with dexamethasone, a synthetic glucocorticoid, injected daily during the last third of pregnancy induces activation of the nucleotide-binding oligomerization domain (NOD)-, leucine-rich repeats (LRR)- and pyrin domain-containing protein 3 (NLRP3) inflammasome, expression of pro-inflammatory cytokines (IL-1β and TNF-α), activation of purinergic receptors, and increase in Panx1 HC activity in hippocampal oligodendrocytes of the offspring [128]. Interestingly, the inhibition of MCs or microglia activation with masitinib and minocycline, respectively, prevents the increase HC activity of oligodendrocytes, suggesting an upstream modulation of both HC types in these cells. Although MCs seem to be activated before microglia, after 30 min of acute exposure to dexamethasone, the HC activity, evaluated by ethidium bromide uptake, increases in microglia, astrocytes, and oligodendrocytes [128].

### 4.3. Amyotrophic Lateral Sclerosis

Amyotrophic lateral sclerosis (ALS) is an untreatable disease characterized by the degeneration of motoneurons. While the majority of patients have a sporadic form of the disease (sALS), about 10% have familial ALS (fALS), which is associated with pathogenic mutations (mut) in genes such as superoxide dismutase (*SOD1*), transactive response DNA-binding protein 43 (*TARDBP* encoding TDP43), and *C9ORF72* (which is characterized by an intronic hexanucleotide expansion) [129,130,131]. Mutations in *C9ORF72* and *TARDBP* are also detected in patients with frontotemporal dementia (FTD), which is the second most common cause of early dementia (<65 years) after AD, which is characterized by progressive deficits in behavior, language, and executive functions, due to a progressive neuronal atrophy and death in the frontal and temporal cortices [132]. Mutations in *C9ORF72* and *TARDBP* are also present in families that suffer simultaneously from both diseases and thus exhibit motor dysfunction as well as cognitive impairments. Collectively, familial and sporadic patients with ALS/FTD share many clinical and histopathological features that suggest the involvement of a convergent, common mechanistic pathway.

Evidence obtained from co-cultures of brain cells as well as from animal models, strongly indicate that astrocytes play a crucial role during the onset and progression of ALS/FTD (reviewed in [133]). Thus, motoneuron damage and cell death is mediated by astrocytes rather than intrinsic pathological processes of the motoneurons, and therefore, they are called non-cell autonomous mechanisms. Non-cell autonomous toxicity to motoneurons has also been shown using conditioned media from human and mouse astrocytes expressing ALS-linked mutations (*SOD1*, *TARDBP, C9ORF72*) or that lack identified causes of ALS (sALS) [134,135,136,137,138,139,140,141,142]. Despite considerable efforts, so far, the toxic factor released by ALS astrocytes has not been identified, although a recent study by the Przedborski laboratory indicates that mutant SOD1 astrocytes release a soluble protein(s) or fragment(s) between 5 and 30 kDa [143]. These and other studies rule out several usual suspect molecules, such as glutamate, ATP, oxygen and nitrogen reactive species, TNFα, and SOD1 [134,137,143]. Nonetheless, astrocytes release numerous additional small pro-inflammatory molecules or bioactive substances, which can have pathological effects on motoneurons (see Figure 2; [144,145]).

Accumulating data implicate a role for HCs in ALS: they contribute to the secretion of various molecules, either directly through their pore, or indirectly by inducing vesicular release (Figure 2; [47,146]). In particular, Cx43 HCs are predominantly expressed in astrocytes throughout the central nervous system (CNS) [150,151], and several studies report that Cx43 (but not Cx30) protein expression is elevated in the spinal cord of symptomatic mutant SOD1 rodent models [152,153,154,155,156]. Cx43 protein expression is also increased in postmortem spinal cord and motor cortex tissues of sALS patients [155]. Moreover, the blockade of Cx43 HCs suppresses the disease progression (neuronal loss at the spinal cord and extended survival) in mutant SOD1 animals treated with INI-0602 [92]. However, the Panx1 HCs contribution should not be discarded, since INI-0602 is derived from glycyrrhetinic acid, which is a molecule that not only inhibits Cxs but also blocks Panx1 HCs [157].

Astrocyte cultures harvested from the spinal cord of mutant SOD1 mice also display increased Cx43 protein expression [155,158]; these cultured ALS astrocytes form functional Cx43 HCs, as determined by ethidium bromide uptake in the presence of the specific Cx43 HC blocker Gap26 [155]. Notably, the Maragakis group has documented that relative to control human astrocytes, Cx43 protein expression is augmented in human-induced pluripotent stem cell (iPSC)-derived astrocytes from sALS (≈5-fold) and fALS patients who harbor mutant SOD1 (≈3-fold) or C9ORF72 (≈6-fold) [155]. Given that such iPSC-derived cultures are highly enriched in astrocytes and lack microglial cells [135], these results infer that Cx43 HCs are intrinsically increased in ALS astrocytes. The application of the Cx43-based channel blockers Gap26 (HCs and GJs) or Gap19 (HCs) to astrocyte–motoneuron co-cultures significantly—but modestly—protects motoneurons from cell death [155]. Together, these results point to a role for Cx43 HCs in the release of the toxic molecule(s) that are present in ALS astrocyte-conditioned media and furthermore suggest that unknown channels and/or receptors also contribute to the release mechanism: potential candidates are Panx1 channels and purinergic P2X and P2Y receptors (see Figure 2). Primary astrocyte cultures of neonatal mutant SOD1 mice display higher amounts of Panx1 but not of P2X7 [158]. Moreover, Panx1 transcripts are increased in the spinal cord of symptomatic mutant SOD1 mice [156], and as discussed above, the treatment of mutant SOD1 animals with the dual Cx and Panx1 HCs blocker INI-0602 was beneficial [92,157]. In addition to their roles in ALS pathogenesis, Cx43 HCs and Panx1 HCs could also serve a critical function in the progression of ALS by driving inflammatory responses (see below). 

Although much less studied relative to ALS/FTD, studies in vitro have shown that astrocytic non-cell autonomous effects are also evident in Alzheimer’s, Parkinson’s, and Huntington’s disease [133]. For example, the release of glutamate and ATP by Alzheimer´s disease astrocytes [68] and α-synuclein by Parkinson’s disease astrocytes [159] contributes to neurodegeneration. For the role of glial Cx HCs and Panx HCs in Parkinson’s disease and Huntington’s disease, we refer the reader to a recent review [47]. 

To our knowledge, direct evidence for MCs activation through an HC-dependent mechanism has not been obtained in ALS. However, increased expression of Cx43 in spinal cord of experimental models of ALS and in patients has been reported [152,153,154,155,156]. In the ventral spinal cord of symptomatic mutant SOD1 mice, immunofluorescent staining also reveals intense Cx43 labeling not only in GFAP-positive astrocytes but also in surrounding GFAP-negative cells [155]. Indeed, Cx43 is expressed by MCs (see Figure 1) and activated microglia. Activation of resident and infiltrated MCs is of particular concern in ALS, as these cells can significantly worsen the pro-inflammatory environment by unleashing a cytokine storm (see Figure 1 and below). Here, we briefly discuss the presence and role of MCs in ALS; we refer the reader to recent reviews for additional information [126,160,161,162]. 

In recent years, the Barbeito group has rigorously demonstrated a significant increase in the number of MCs in a rat model of ALS, including in the extensor digitorum longus (EDL) muscle, at the neuromuscular junction (NMJ), and in the sciatic nerve and its ventral roots, starting at the time of disease onset [163,164,165]. The authors have also reported an increased number of MCs in postmortem skeletal muscle of sALS patients [164]. In this work, MCs were identified by their strong immunoreactivity and co-expression of tryptase, chymase, and tyrosine kinase receptor c-kit, while additionally using Toluidine blue staining to identify a MC degranulating phenotype in the muscles of animals and patients with ALS. Even though resident MCs are also observed in wild-type rats and healthy subjects, these cells showed no evidence of degranulation. Together, these results suggest that MCs infiltrate and/or proliferate in the skeletal muscle of both an ALS experimental model and in patients with ALS, and that their degranulation contributes to disease progression. 

Promising treatment of symptomatic mutant SOD1 rats with the tyrosine kinase inhibitor masitinib has been shown to decrease the number of MCs (and neutrophils) in the EDL muscle, and it reduces the rate of NMJ denervation and motor deficits in rats bearing the mutant SOD1 [163,166]. Clinical tests of masitinib in a phase III ALS trial also demonstrate therapeutic benefits by slowing the deterioration of motor functions [119]. It remains unclear how exactly masitinib targets skeletal muscle MCs in ALS and if this drug prevents their activation and/or accumulation (through infiltration and/or local proliferation); studies with human cord-blood-derived MCs show that masitinib inhibits migration and degranulation [114]. It is also not known whether MCs in the spinal cord play a role in ALS and whether they are specifically targeted by masitinib, as immunofluorescence assays fail to detect MCs in the spinal cord of symptomatic ALS rats [164] and mice [167]. This is a surprising finding, because MCs, as with other bone marrow-derived immune cells, would be expected to infiltrate the brain and spinal cord upon the disruption of the BBB. Sparse accumulation of tryptase-positive MCs has been reported in postmortem spinal cord tissue of patients, while no MCs have been detected in control subjects [168,169]. These latter results support findings in healthy tissue that resident MCs are located in the dura mater of the meninges around the spinal cord but not in the parenchyma [170,171,172]. It would be of interest to understand why MCs accumulation is different in rodent versus human spinal cord in ALS. 

While masitinib likely delays disease progression in ALS patients by targeting MCs, this inhibitor of the type-3 tyrosine kinase receptor also targets colony-stimulating factor 1 receptors (CSF-1Rs) that are expressed by microglial cells, pointing to a double action. Similar neuroprotective effects in ALS were also observed in another study that used a compound that targets MCs as well as microglia. Thus, chronic treatment of mutant SOD1 mice (from postnatal day 60 onwards) with diosodium chromoglycate (cromolyn)—an FDA-approved compound that inhibits MC activation and degranulation and induces anti-inflammatory microglia activation—decreases the degranulation of MCs in the anterior tibialis muscle, reduces NMJ denervation, and extends the survival of ALS mice [173]. Cromolyn treatment also decreases the levels of some pro-inflammatory cytokines/chemokines, including TNFα [162,173].

## 5. Conclusions and Future Directions

The outstanding plasticity of MCs is displayed upon a variety of stimuli, which can lead to different activation responses: (1) release of pre-stored inflammatory mediators, (2) release of de novo synthetized inflammatory mediators, or (3) both processes. These responses impact the microenvironment of the tissue, since they cause vascular permeabilization, leukocyte infiltration, and parenchymal cell interactions, among other outcomes. Remarkably, these differences in activation response are evident among members of a heterogeneous MC population. Conversely, at least in two different populations of murine MCs, immature bone marrow and mature brain MCs, Cx43, and Panx1 HCs opening seem to mediate a key role during acute degranulation responses (including amyloid peptide activation and crosslinking recognition). Despite the sharing progress so far generated, important questions regarding MC biology still remain to be answered; here, we briefly discuss several of them.

Particularly for the crosslinking IgE–FcεRI response, the intracellular pathways and cargo release have been extensively studied during the past 50 years. However, mediators and cellular events triggered after complement fragments, viral particles, or neuropeptides treatment are still poorly defined. It would be of interest to evaluate whether Cx/Panx HCs and purinergic signaling are also involved in MC activation in the absence of degranulation. In addition, many of the responses of activated MCs have been evaluated on acute conditions. However, they remain poorly studied during chronic conditions (e.g., AD and ALS). In addition, it remains unknown whether under chronic conditions there is a persistent release of granule content or a selective constitutive secretion. For the AD context, how MCs interact with glia and neurons remains unknown. Other relevant questions are: Is the increase in brain MCs number explained by MCs infiltration and/or proliferation? So far, the involvement of Panx1 and Cx43 HCs has been evaluated in the APP/PS1 mice model and mainly using in vitro assays. Whether the MC population response is similar in non-transgenic models of AD such as *Octodon degus*, non-human primates, or dogs would provide further proof of concept.

Similarly, plenty of literature support an important contribution of MCs during stress conditions. However, the molecular identity of MC HCs present in chronic or acute stress models has not been established, and the development of selective and potent blockers of different HC types is of great interest not only to further understand their functional role, but it would also be useful to reduce or prevent MC activation in different brain diseases that evolve with the relevant neuroinflammatory response. In the same line of thought, in vitro and in vivo models indicate a central role for astrocyte HCs during the onset of ALS with an important role of degranulating MCs during the progression of this disease. Thus, therapy designed to prevent or reduce MC activation in human ALS patients is promising. In this context, how MCs become activated and communicate with astrocytes still needs to be clarified. 

Regarding other neurodegenerative conditions, including Huntington´s and Parkinson´s disease, the role of MCs HCs activation and their glial cell interaction are poorly studied, and the current knowledge in AD and ALS can be useful to study these diseases. 

## Figures and Tables

**Figure 1 ijms-22-01924-f001:**
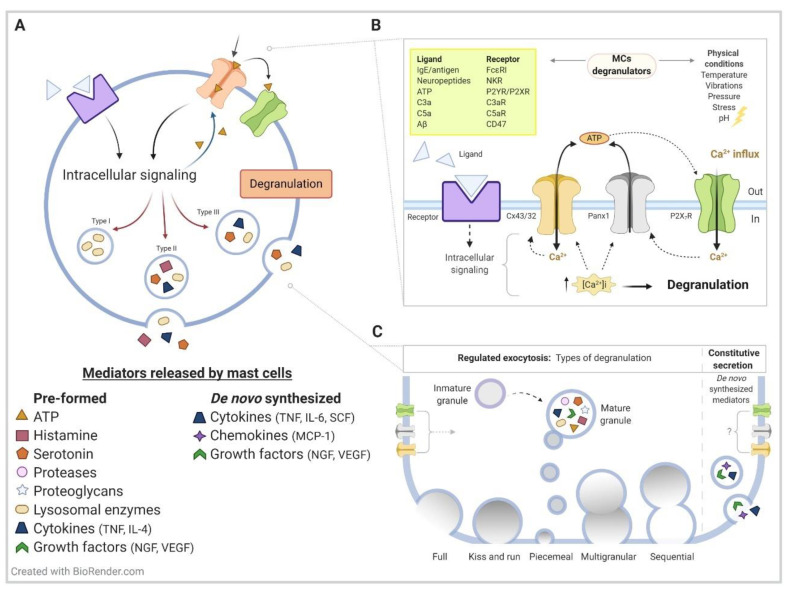
General mechanisms of degranulation from activated MCs**.** (**A**) Mast cells (MCs) are resident immune cell characterized by presenting numerous secretory granules in their cytoplasm in which they store pre-formed inflammatory mediators, such as biogenic amines (e.g., histamine, serotonin), growth factors (e.g., nerve growth factor (NGF), vascular endothelial growth factor (VEGF)), specific proteases (e.g., tryptase, chymase, carboxypeptidase A3), serglycin proteoglycan (such as heparin and chondroitin sulfate), cytokines (e.g., IL-4 and tumor necrosis factor (TNFα)), and adenosine triphosphate (ATP). These mediators are differentially packed in at least three different types (Types I, II, and III) of heterogenous secretory granules along the cell. Upon activation, these pre-formed mediators are rapidly released to the extracellular medium in a process called degranulation or regulated exocytosis, leading to immediate inflammatory reaction. Depending on the stimulus, MCs activation can also trigger the de novo synthesis of inflammatory mediators in a process called constitutive secretion, releasing molecules such as neuropeptides (e.g., substance P), growth factors (e.g., NGF, VEGF, platelet derived growth factor (PDGF)), cytokines (e.g., IL-6, IL-1β, TNFα, SCF), and chemokines (e.g., Monocyte chemoattractant protein-1 (MCP-1)), modulating late phase inflammation responses. Both docking and fusion to the plasma membrane are dependent on increased levels of the Ca^2+^ signal. The intracellular events triggered after MCs activation includes increasing membrane permeability to Ca^2+^, ATP release, and purinergic receptor recruitment, which are key components for MCs degranulation. (**B**) Different agents can induce MCs degranulation, including physical changes (e.g., temperature, vibrations, pressure, stress, pH) and ligand–receptor interaction (e.g., antigen–Immunoglobulin E (IgE)–FcεRI, ATP–P2Y/X receptors, and Aβ peptide–CD47). For the latter case, after intracellular signals are generated, connexins (Cxs) (potentially mediated by Cx43 and/or Cx32) and Panx1 HCs are eventually activated. HCs opening leads to massive ATP release, activating adjacent purinergic receptors in an autocrine way. Purinergic P2X receptor activation allows Ca^2+^ influx and the direct recruitment of more Panx1 HCs opening, leading to a vicious loop resulting in degranulation. (**C**) Different types of degranulation have been reported for MCs, including full exocytosis, kiss-and-run exocytosis, piecemeal degranulation, multigranular, and sequential compound exocytosis. However, compound exocytosis and piecemeal degranulation are the prevailing forms of degranulation in human, mouse, and rat MCs. Although we have proposed HCs and purinergic receptors activation as key components for MCs degranulation, contribution on constitutive secretion have not been evaluated. For more information about inflammatory mediators, we recommend [10,20]; while for detailed reviews of regulated exocytosis mechanism, we recommend revising [13,14,21,22,23]. Finally, the evidence of hemichannels (HCs) contribution on MCs degranulation is indicated in [24,25].

**Figure 2 ijms-22-01924-f002:**
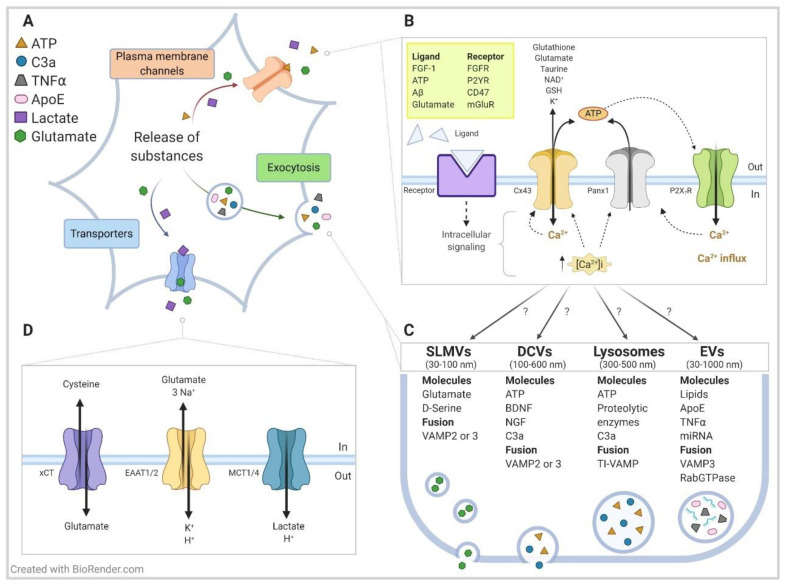
Mechanisms of storage and release of bioactive substances in astrocytes: a focus on neurodegenerative diseases. Model showing (**A**) a general overview and (**B–D**) specific details of the release of signaling molecules from astrocytes into the extracellular space through three classical mechanisms. These mechanisms involve (**B**) diffusion through plasma membrane channels, (**C**) exocytosis, and (**D**) translocation by transporters. For simplicity, we show only a few membrane proteins, along with some bioactive substances and secretion signaling mechanisms that are critical for communicating with neurons and glial cells and are involved in diverse neurodegenerative diseases, particularly in Alzheimer´s disease and amyotrophic lateral sclerosis (ALS)/frontotemporal dementia (FTD). (**B**) Plasma membrane channels/receptors include HCs (mainly Cx43 and Panx1), purinergic receptors (e.g., P2XR, P2YR), growth factor receptors (e.g., fibroblast growth factor receptor (FGFR)) and ionic channels (e.g., VRAC, TREK-1; not shown). The influx of Ca^2+^ through HCs and ion channels—along with the initiation of intracellular signaling cascades through purinergic and growth factor receptors—promotes the release of Ca^2+^ from intracellular stores (predominantly endoplasmic reticulum) and increases cytoplasmic Ca^2+^ concentration ([Ca^2+^]_i_), which is a process that leads to the extracellular release of many bioactive substances by exocytosis. [Ca^2+^]_i_ promotes the opening of Cx43 HCs and Panx HCs to directly release ATP into the extracellular environment. Cx43 HCs also release other small substances such as glutamate, NAD^+^, and glutathione. Extracellular Aβ also activates HCs. (**C**) Vesicles, such as SLMVs (synaptic-like microvesicles), DCVs (dense-core vesicles), lysosomes and EVs (extracellular vesicles, mainly microvesicles) contain diverse bioactive substances, including neurotransmitters, hormones and peptides, metabolic substrates, growth factors, and inflammatory factors, among others. Particular membrane fusion molecules (e.g., VAMP2 and TiVAMP) regulate the release of the vesicles. While many membrane proteins and bioactive substances are identified, little is known about the spatial and temporal aspects of vesicle release, including if membrane receptors/channels are localized in specific microdomains to trigger localized intracellular signaling pathways, and if elevated, [Ca^2+^]_i_ promotes the fusion of specific vesicle types with a membrane that is adjacent either to neuronal synapses or away from the tripartite synapse. Understanding the mechanisms that underlie the differential release of vesicles would potentially open therapeutic avenues for constraining the secretion of critical toxic factors—such as glutamate, ATP, C3a, TNFα, apoliprotein E (ApoE), and certain miRNAs identified to play a relevant role in the pathogenesis of Alzheimer´s disease and ALS/FTD—without affecting the release of beneficial factors. Although less intensively studied, several of these toxic factors have been implicated in other neurodegenerative diseases. Of interest, a recent report shows that mutant huntingtin (mHtt) associates with Rab3a, which is a small GTPase localized on the membranes of DCVs, and it impairs brain-derived neurotrophic factor (BDNF) release from astrocytes. Another emerging topic is the potential role of connexons (mediated by Cx43 and Cx26) in recruiting molecules such as RNA and DNA into microvesicles and exosomes and the subsequent intercellular transfer of these vesicles. For simplicity, mHtt and connexons are not shown in the model. (**D**) The critical bioactive substances glutamate and lactate can also be released by transmembrane transporters such as the cystiene/glutamate antiporter (xCT), and monocarboxylate transporters (MCT1/4), respectively, independent of [Ca^2+^]_i_ changes. The Na^+^-dependent glutamate excitatory amino acid transporters (EAAT1/2) are relevant for the uptake of extracellular glutamate and make a well-established contribution to excitotoxicity in ALS. For additional information on the release of bioactive substances and on the role of HCs in astrocytes, see [47,144,145,146,147]; on the Aβ-mediated opening of HCs, see [68]; on mHtt and impaired BDNF release from astrocytes, see [148]; and on connexons in the intercellular transfer of vesicles, see [149].

## Data Availability

Not applicable.

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
