# Peer review of "Mast Cell and Astrocyte Hemichannels and Their Role in Alzheimer’s Disease, ALS, and Harmful Stress Conditions"

_ijms, 2021, doi:10.3390/ijms22041924_

Round 1

Reviewer 1 Report

The review by Paloma A. Harcha et al. entitled “Mast cells hemichannels and their interactions during neuroinflammatory diseases” focus on hemichannels and their contribution in MC degranulation in AD, ALS and harmful stress conditions.

The review is original and of interest in its field and is clearly written.

 I recommend that the paper be accepted with minor revision:

  1. Please provide the full name of all acronyms the first time they are mentioned.
  2. The authors should better empathize the title
  3. The authors should better describe the conclusions.
  4. The literature is poorly updated. Please add recent references.

Author Response

We thank the reviewer for evaluating our work and we are pleased with his/her appreciation of the originality of our review. We include below responses on the specific comments/suggestions made. 

  1. Please provide the full name of all acronyms the first time they are mentioned.

Thanks for pointing it out, we carefully revised the text and provided the full name of all acronyms the first time mentioned. Including:  

28 adenosine triphosphate

29 tumor necrosis factor, apoliprotein E

61 Immunoglobulin E

66 tumor necrosis factor, reactive oxygen species

77 calcitonin gene-related peptide, corticotropin-releasing hormone

83 lipopolysaccharides, polyinosinic:polycytidylic acid

106 calcium ion

132 Phospholipase C

133 phosphatidylinositol 4,5-biphosphate, inositol 1,4,5-trisphosphate, diacylglycerol

134 protein kinase C

136 stromal interaction molecule 1

137 calcium release-activated calcium channel protein 1

138 transient receptor potential channel

141 ovalbumin

142 ionotropic purinergic

166 glial fibrillary acidic protein

174 N-methyl-D-aspartate

197 interleukin 1

219 vascular endothelial growth factor

273 NOD-, LRR- and pyrin domain-containing protein 3

  1. The authors should better empathize the title

We changed the title as suggested to “Mast cell and astrocyte hemichannels and their role in Alzheimer's disease, ALS and harmful stress conditions” (line 2-3)

  1. The authors should better describe the conclusions.

We added it in a new section at the end “Conclusions and future directions”

  1. The literature is poorly updated. Please add recent references.

We indeed cited updated literature (1972-1990) but that correspond to the original pioneer work of MCs granule characterization and their distribution along the central nervous system, but the rest of the literature is updated.

Reviewer 2 Report

The authors described here that hemichannel activation in mast cells might be involved in the pathology of neuroinflammatory diseases, such as Alzheimer's diseases and amyotrophic lateral sclerosis. Because this topic is quite unique and original, this review manuscript should be appealing to the researchers in this field. However, because the number of the closely related published papers is limited, it might be of significance for the readers if it is more concisely written.

1. The authors should summarize the basic knowledge of 1) mast cells and 2) hemichannels at the beginning of this review and then discuss the roles of hemichannel activation in mast cells.

2. A measurable part of this review is composed of the findings of hemichannel activation of various non-mast cells in the neuroinflammatory diseases. The authors should focus on mast cells and discuss the significance of hemichannel activation in mast cells in comparison of that in the other cells, such as neuron, astrocyte, and microglia. They could delete a large part of the section of ALS, because it describes the roles of hemichannels of non-mast cells and the roles of mast cells in ALS may be independent of the hemichannel activity of mast cells. Box 2 summarizes the roles of astrocytes in neuroinflammation. If the authors would like to include such contents here, it might be better to change the title.

3. The reference #74 should be discussed in detail here, but the authors seems to depend on this report too heavily. The findings reported in the reference #74 should be verified by citing the other related study.

4. The concept of heterogeneity of cytoplasmic granules of mast cells might not be generally accepted and the authors should cite the suitable papers to support it.

5. The figures in Boxes should be revised to clearly show the difference of granule contents. It is quite difficult to distinguish the colors of the symbols.

Author Response

We thank the reviewer for evaluating our work and we are pleased with his/her appreciation of the originality and uniqueness of our review. We include below responses on the specific comments/suggestions made. 

  1. The authors should summarize the basic knowledge of 1) mast cells and 2) hemichannels at the beginning of this review and then discuss the roles of hemichannel activation in mast cells.

We changed the order of the review and put the description of mast cells under “Mast cell generalities” (lines 41-85). The hemichannel section is now put immediately after the mast cell section under “Hemichannels overview (line 86-123).

  1. A measurable part of this review is composed of the findings of hemichannel activation of various non-mast cells in the neuroinflammatory diseases. The authors should focus on mast cells and discuss the significance of hemichannel activation in mast cells in comparison of that in the other cells, such as neuron, astrocyte, and microglia. They could delete a large part of the section of ALS, because it describes the roles of hemichannels of non-mast cells and the roles of mast cells in ALS may be independent of the hemichannel activity of mast cells. Box 2 summarizes the roles of astrocytes in neuroinflammation. If the authors would like to include such contents here, it might be better to change the title.

We changed title of the review to “Mast cell and astrocyte hemichannels and their role in Alzheimer's disease, ALS and harmful stress conditions” (lines 2-3). Additionally, we eliminated the sections related to C3 activated astrocytes in ALS (lines 396-444).

  1. The reference #74 should be discussed in detail here, but the authors seems to depend on this report too heavily. The findings reported in the reference #74 should be verified by citing the other related study.

We strongly focus on reference #74 since is the first work that evaluate HCs activity on MCs (particularly on an AD context), only supported later by ref #38 (allergy model) and 121 (prenatal stress). Sadly, to our knowledge, there are no other related studies to validate our work.

  1. The concept of heterogeneity of cytoplasmic granules of mast cells might not be generally accepted and the authors should cite the suitable papers to support it.

We include suitable papers as suggested, particularly work from Dvorak AM 2005; Raposo G 1997; Puri N 2008; Moon 2014 (line 73)

  1. The figures in Boxes should be revised to clearly show the difference of granule contents. It is quite difficult to distinguish the colors of the symbols.

 We changed the colors and size of granule contents in Box 1 and Box 2.

Round 2

Reviewer 2 Report

I found that the authors adequately responded to the concerns that I raised. Their focus is clearly presented in the revised manuscript. This important review shed a light on the roles of hemichannels in mast cells. I would like to add some typos as follows.

lines 66, 67, and 68
Parentheses should be used for these abbreviations.